# The Effect of Combined Isometric and Plyometric Training versus Contrast Strength Training on Physical Performance in Male Junior Handball Players

Hédi Allégue [1], Olfa Turki [1,2], Dustin J. Oranchuk [3], Aymen Khemiri [1,2], René Schwesig [4,*,†] and Mohamed Souhaiel Chelly [1,2,†]

[1] Research Laboratory (LR23JS01) Sport Performance, Health & Society, Higher Institute of Sport and Physical Education of Ksar Saîd, University of Manouba, Tunis 1000, Tunisia; hediallegue@gmail.com (H.A.); olfa.turki17@gmail.com (O.T.); aymenkha3@gmail.com (A.K.); mohamedsouhaiel.chelly@issep.uma.tn (M.S.C.)

[2] Higher Institute of Sport and Physical Education of Ksar Saîd, University of Manouba, Tunis 1000, Tunisia

[3] Muscle Morphology, Mechanics, and Performance Laboratory, Department of Physical Medicine and Rehabilitation, University of Colorado Denver-Anschutz Medical Campus, Aurora, CO 80045, USA; dustinoranchuk@gmail.com

[4] Department of Orthopedic and Trauma Surgery, Martin-Luther-University Halle-Wittenberg, Ernst-Grube-Str. 40, 06120 Halle, Germany

[*] Correspondence: rene.schwesig@uk-halle.de

[†] The last two authors are co-last authors.

**Featured Application: The purpose of the present study was to compare the effects of 8 weeks of contrast strength training versus combined isometric and plyometric training on sprinting, change of direction, throwing, and handgrip strength. The combined approach yielded greater enhancements in sprinting and ball throwing velocity compared to the contrast group. Both training strategies significantly improved most neuromuscular performance measures compared to the control group.**

**Abstract:** Exploring resistance training methods is crucial for optimizing performance programs. Isometric muscle actions have gained popularity in athletic training, but their impact on dynamic performance is uncertain. Isolated isometric actions also lack ecological validity. We compared the effects of 8-week combined isometric and plyometric (COMB) training and contrast strength training (CST) programs on junior male handball players. Thirty-six male first national division players ($17.6 \pm 1.0$ years) were enrolled and randomly assigned to COMB, CST, or control (CONT) groups (all n = 12). Sprinting, change of direction, ball throwing velocity, jumping, and strength were assessed pre- and post-intervention. A significant group × time interaction was observed between the COMB and CONT groups for 20 and 30 m sprints ($p \leq 0.002$) and between the COMB and CST groups ($p \leq 0.042$). The COMB group had the largest improvements in change of direction and the modified T-test, with significant group × time interactions between the COMB and CONT groups ($p \leq 0.021$). Significant group × time interactions were observed between the COMB and CST groups and between the COMB and CONT groups for 3 step running throw ($p = 0.003$; $p < 0.001$), running throw ($p = 0.02$; $p = 0.031$), and jumping throw ($p = 0.001$; $p < 0.001$). Countermovement jump showed a significant group × time interaction ($p = 0.014$), with the COMB group outperforming the other groups. Generally, COMB yielded larger improvements than CST. Coaches should consider incorporating a combination of isometric and plyometric exercises for in-season strength training.

**Keywords:** agility; vertical jump; sprinting; change of direction; throwing; strength; youth athletes

## 1. Introduction

Handball demands high-intensity, short-duration physical effort, strenuous contact, and explosive muscular contractions [1,2]. As such, handball players are required to engage in frequent bouts of high-intensity activities such as sprinting, jumping, throwing, and quick changes in direction [2–5] as well as physical contact to gain an advantageous position for the throwing player [5–7]. It is therefore advisable that handball athletes incorporate specialized handball conditioning programs that prioritize high-intensity exercises.

The effectiveness of strength and power training programs in enhancing athletic capabilities among team sport players has been consistently demonstrated [8–10]. These programs elicit positive adaptations that can be attributed to various neuromuscular aspects. Physiological mechanisms such as the storage and utilization of elastic energy and the function of the stretch-shortening cycle are known to change [11]. Additionally, morphological factors, including muscle architecture and fiber type, and neural factors like motor unit recruitment, synchronization, firing frequency, and intermuscular coordination also contribute to these adaptations [12]. Another key factor that contributes to the observed improvements is the increased rate of force development, which refers to the maximal rate at which muscle force rises during the initial phase of a muscle contraction [13,14]. These factors could explain the enhancements in physical performance following participation in strength and power training programs.

Training strategies such as contrast [15] and plyometric [3,16] training methods are widely used techniques among individuals engaged in dynamic sports to improve the dynamic muscular performance of team sport players. Previous studies have demonstrated that both dynamic heavy resistance training involving low-velocity movements and plyometric training involving high-velocity movements can independently enhance power and rate of force development [13,17,18]. A recent review described the contrast strength training (CT) method as an exercise sequence alternating high- and low-load (higher velocity) exercises in a set-by-set fashion within the same session (corresponding to 'contrast pairs' and 'intra-contrast rest') [15]. Several studies have demonstrated the efficacy of contrast training in enhancing strength [19,20], power [16], jumping [19,20], sprinting [19,21], agility, and repeated change of direction performance [21]. Contrast strength training is also a promising strategy for improving maximal strength and power in junior team sport players [16,18]. Moreover, a specific contrast strength training program without external loads (12 weeks of a combined isometric + plyometric training program) was suggested for young soccer players as an effective training strategy to improve soccer-specific skills such as vertical jumping, sprinting, agility, and kicking speed [21].

Along with the traditional and contrast strength training methods, the isometric training approach has been used in athletes' physical preparation processes. Isometric muscle actions involve contracting the skeletal muscles without external movement, enabling precise force application control at specific joint angles [22]. Isometrics are often considered less advantageous for sports performance due to their static nature [22]. However, recent findings indicate that isometric training might offer several advantages over dynamic strength training, such as lower energy expenditure [23,24], greater improvements in tendon stiffness during progressive and ballistic contractions [22,25], and angle-specific strength [22,26,27]. Therefore, Lum and Barbosa [22] recommended incorporating isometric strength training alongside dynamic exercises to fully optimize the benefits for activities encompassing all three phases of the stretch-shortening cycle, such as countermovement jumping.

Understanding how combining more than one training modality affects the development of sport-related performance is critical for coaches and resistance training practitioners to design more efficient training programs. For handball players, enhancing the quality of training is essential to maximize their performance during training and competition. Moreover, to our knowledge, no previous study has previously tested a training protocol that combined isometric and plyometric drills in team sport players. In this regard, comparing contrast strength training and combined training (strength "isometric work" and dynamic) may help assess which modality is more effective in optimizing players' physical

performance. Therefore, the purpose of the present study was to compare the effects of 8 weeks of contrast strength training versus combined isometric and plyometric training on sprinting, change of direction, throwing, and handgrip strength. We hypothesized that the contrast training and combined isometric and plyometric training programs would substantially improve dynamic performance compared to the control group.

## 2. Materials and Methods

### 2.1. Participants

Thirty-six participants classified as highly trained were recruited from the same male handball team in the first national (elite) division [28]. All players were starters for their teams and had at least 12 years of experience. All measurements were performed in the middle of the season, and the subjects were already trained and accustomed to muscle-strengthening programs (2 months of post-season physical preparation). None of the participants suffered from injuries or diseases during the training period. Throughout the test period, there were no dropouts among the participants. Participants were randomly allocated to the combined isometric and plyometric training (COMB) group (n = 12), the contrast strength training (CST) group (n = 12), or the control (CONT) group (n = 12) (Table 1). For age ($\eta_p^2 = 0.626$) and peak height velocity ($\eta_p^2 = 0.330$), relevant mean differences between groups were observed, especially compared with the COMB group (Table 1).

**Table 1.** Demographic and anthropometric characteristics of different groups. Values are given as mean ± standard deviation. Relevant main group effects are marked in bold.

| Parameters | Groups | | | Variance Analysis | |
|---|---|---|---|---|---|
| | | | | Main Group Effect | Partial Effects |
| | COMB (n = 12) | CST (n = 12) | CONT (n = 12) | $p/\eta_p^2$ | (p) |
| Age (years) | 18.6 ± 0.24 | 16.7 ± 0.40 | 17.6 ± 0.99 | **<0.001/0.626** | COMB vs. CST: <0.001 CST vs. GC: 0.002 COMB vs. GC: 0.003 |
| Height (cm) | 181.1 ± 5.87 | 179.1 ± 7.88 | 180.1 ± 3.78 | 0.725/0.019 | - |
| Weight (kg) | 76.3 ± 8.81 | 70.0 ± 12.3 | 75.3 ± 11.0 | 0.322/0.066 | - |
| Body Fat (%) | 14.5 ± 3.98 | 13.5 ± 8.34 | 17.3 ± 7.35 | 0.368/0.059 | - |
| Peak h = Height Velocity (years) | 2.79 ± 0.71 | 1.52 ± 0.70 | 2.07 ± 0.90 | **0.001/0.330** | COMB vs. CST: <0.001 |

Throughout the investigation, both groups trained with 5 weekly training sessions, and an official match was played on Sundays. During the investigation, participants in the control group continued their usual handball training program without any resistance training. The standard training sessions, lasting 90 to 100 min, included technical and tactical activities at various intensities, along with 25 to 30 min of continuous play. All participants and their legal representatives were informed about all testing and training procedures and the potential benefits and harms related to the study. Verbal and written informed consent (legal representatives) and assent (children) were obtained before the start of the experiment. The local Institutional Review Committee of the Higher Institute of Sport and Physical Education of Ksar-Saîd, Tunisia, approved all procedures (LR23JS01). All procedures were performed in accordance with the latest version of the Declaration of Helsinki.

### 2.2. Anthropometrics

Body height and mass were assessed using a stadiometer and weighing scales. The overall percentage of body fat was estimated from the biceps, triceps, subscapularis, and supra iliac skinfolds, using the equation of Durnin and Womersely for adolescent males aged 16–19.9 years [29]:

$$\% \text{ Body fat} = (4.95/(\text{Density} - 4.5)) \times 100$$

where Density = 1.162–0.063 (LOG sum of 4 skinfolds).

### 2.3. Linear Sprint Test

Linear sprint performance was evaluated using a maximal 30 m sprint with split times at 5 m and 20 m [30]. Participants were instructed to run as fast as possible along the 30 m distance from a standing start, to not slow down before passing the finish line, to start the sprint when ready from a standing start position, and to place their front foot behind a land marker placed 20 cm before the first timing gate. Time performance was recorded using photocell gates (Witty GATE 2.0.8, Microgate, Italy) with a measure precession of 0.01 s, placed 0.5 m above the ground. Subjects performed two trials with 2 min of rest between trials. The best trial was selected for further analysis.

### 2.4. Modified T-Test

Based on a previous protocol [31], the participants were instructed to begin with both feet behind the starting line (cone A). No starting command was given, and participants were instructed to start of their own volition. Participants were instructed to sprint forward to cone B and touch its base with the right hand. Then, facing forward and without crossing their feet, shuffle left to cone C and touch its base with the left hand. Then, shuffle right to cone D and touch its base with the right hand, then shuffle back to the left to cone B and touch its base with the left hand. Finally, backpedal as quickly as possible and return to the starting line. Time was measured using photocell electronic timing sensors (Racetime2 SF Kit, Microgate, Bolzano, Italy), which were placed 75 cm above the ground, 3 m apart, and facing each other at the starting line (A).

### 2.5. Repeated Change of Direction Test

The repeated change of direction (RCOD) test consisted of 6 sprints of 20 m, each starting from a standing position, 0.2 m behind the first gate, with active recovery intervals of 25 s [32]. Time was measured using infrared cells (Witty GATE 2.0.8, Microgate, Italy) located 0.5 m above the ground at the start and finish lines. Four $100°$ direction changes were made at 4 m intervals. During the active recovery phase, the participants were instructed to walk slowly back to the start line. The best time in a single trial (RCOD-Best), the average time for the $6 \times 20$ m sprints (RCOD-Mean), and the total time for the 6 sprint repetitions (RCOD-Total Time) were recorded. The RCOD-Fatigue Index was calculated using the formula:

$$\text{RCOD-Fatigue Index} = 100 \times (\text{RCOD-Total Time}/\text{ideal sprint time}) - 100$$

where Ideal sprint time = number of sprints $\times$ RCOD-Best.

An adapted COD deficit calculation was used to evaluate the efficacy of each athlete's ability to utilize their linear speed during a specific change of direction (COD) task, as previously described [33]. Thus, the COD deficit was calculated as follows:

$$20 \text{ m linear sprint time performance} - \text{RCOD-Best time performance}$$

### 2.6. Ball Throwing Velocity Tests

Four types of overarm throws [3] were performed on an indoor handball court: a 3 step running throw, a running throw, a jumping throw, and a standing (penalty) throw. The ball throwing velocity was measured using a radar Stalker ATS II system™ (Radar Sales, Minneapolis, MN, USA) hand-held at shoulder level. The maximal ball velocity was noted for three consecutive trials for each throw type, each separated by at least 15 s of recovery. Players were immediately informed of their performance to maximize motivation, and the fastest of their three values was recorded.

## 2.7. Handgrip Strength Test

A hand dynamometer (Takei, Tokyo, Japan) was held with the arm at right angles and the elbows at the side of the body. The instrument was adjusted so that its base rested on the first metacarpal, and the handle rested on the middle of the participant's four fingers. A maximal isometric effort was maintained for 5 s without ancillary body movement. Two trials were performed with each hand, with 1 min of rest between trials, and the highest reading was used in subsequent analysis.

## 2.8. Back Extensor Strength Test

Maximal isometric back extensor strength was measured in kilograms using a back and leg dynamometer (Takei, Tokyo, Japan), as previously described [34]. Participants stood on the dynamometer foot stand with their feet shoulder width apart and gripped the handlebar positioned across their thighs. The chain length of the dynamometer was adjusted so that initially the legs were fully extended and the hips were flexed at a 30° angle, positioning the bar at the level of the patella. Participants then stood upright without bending their knees and lifted the dynamometer chain, pulling upward as strongly as possible. Three trials were completed, and the highest score was recorded. A 30 s rest interval was allowed between each trial. Trials were terminated early and repeated if excessive spinal flexion was noticed by the tester.

## 2.9. Vertical Jumping Tests

The squat jump (SJ) and countermovement jump (CMJ) were performed on an optical measurement system consisting of a transmitting and receiving bar (Optojump Next, Microgate, Italy). This made it possible to measure flight and contact times during the performance of a series of jumps with an accuracy of 1/1000 s. Dedicated software was used to obtain a series of parameters connected to the athlete's performance with maximum accuracy and in real time. To avoid artificially inflating flight time, participants were instructed to land with their legs fully extended and then to flex their limbs on landing. Participants began the SJ at a knee angle of 90° and, while avoiding any downward movement, they performed a vertical jump by pushing upward. The CMJ began from an upright position, with participants making a rapid downward movement to a knee angle of approximately 90°, arms akimbo, and simultaneously beginning to push off after being instructed to jump as fast and high as possible.

All used tests and parameters showed high to excellent intra-rater reliability and percentage of variation (Table 2).

**Table 2.** Interclass correlation coefficient and coefficient of variation values showing acceptable reliability for running, jumping, throwing, and upper and lower limb force tests.

| Variables | ICC | % CV |
|---|---|---|
| 5 m | 0.96 | 1.4 |
| 20 m | 0.95 | 1.2 |
| 30 m | 0.96 | 1.1 |
| Modified T-Test | 0.94 | 1.2 |
| Squat Jump | 0.96 | 3.0 |
| Countermovement Jump | 0.98 | 2.9 |
| 3 step Running Throw | 0.93 | 1.2 |
| Running Throw | 0.88 | 1.1 |
| Jumping Throw | 0.87 | 1.2 |
| Penalty Throw | 0.96 | 1.2 |
| Handgrip Strength-Right hand | 0.98 | 2.8 |
| Handgrip Strength-Left Hand | 0.98 | 3.5 |
| Back Extensor Isometric Strength | 0.96 | 3.2 |

ICC = interclass correlation coefficient; CV = coefficient of variation.

### 2.10. Training Protocols

The CST and COMB protocols were performed twice weekly (Tuesdays and Thursdays) for eight consecutive weeks. Sessions lasted for 40 min, including a 15 min warm-up.

#### 2.10.1. The Combined Isometric and Plyometric Training Protocol

The COM training program was divided into four stations. Each station included three exercises (4 sets with 1–2 min of rest). The participants were instructed to perform an isometric, a plyometric, and a speed exercise, which could be either a sprint or throw depending on the session. The stations were completed four times during each session, with all sets separated by 120 s of passive rest. The isometric exercise was followed immediately by a plyometric exercise. The quantification of workload is described in Table 3.

**Table 3.** The training program assigned for the isometric training group.

|  | Week | % 1RM | Set × Contraction Time (s) | Recovery Time (min) |
|---|---|---|---|---|
| Cycle 1 | 1 | 60 | 6 × 40 | 1–2 |
|  | 2 | 65 | 6 × 40 | 1–2 |
|  | 3 | 70 | 6 × 40 | 1–2 |
|  | 4 | 70 | 6 × 40 | 1–2 |
| Cycle 2 | 5 | 75 | 6 × 40 | 1–2 |
|  | 6 | 75 | 6 × 50 | 1–2 |
|  | 7 | 75 | 6 × 55 | 1–2 |
|  | 8 | 75 | 3 × 60 | 1–2 |

Station 1: (a) Isometric barbell half squat with a knee angle of 90°. (For the first four weeks, the athlete maintained an isometric hold for 40 s for each exercise, then the isometric duration progressively increased by 5 s each week up to 60 s.); (b) 6 repeated jumps over 30 cm hurdles; (c) 15 m sprint with changes of direction.

Station 2: (a) Isometric barbell bench press with an elbow angle of 90°; (b) 6 horizontal throws of a 2 kg medicine ball located at chest level toward the wall; (c) 4 shots with a handball ball (3 step running throw).

Station 3: (a) Isometric dumbbell Bulgarian split squat (leg on the bench and support leg with a knee angle of 90°); (b) 6 hurdle steps between 25 cm in height (3 on each leg); (c) 15 m sprint with change of direction.

Station 4: (a) Isometric dumbbell pull-over with a trunk–arm angle of 130°; (b) Throw-in with a 2 kg medicine ball; (c) 4 shots with a handball in support.

#### 2.10.2. The Contrast Strength Training Protocol

The CST loads ranged from heavy (60–75% of one repetition maximum [1RM]) to light (30–45%). Each training session included four exercises that were executed in 4 stations: (1) half squat; (2) bench press; (3) Bulgarian split squat, and (4) pull-over. The participants began with a heavy load set that was immediately followed by a set with a low load. All sets were separated by 240 s of passive recovery. Sets, repetitions, rest times, and workload quantification are provided in Table 4.

### 2.11. Statistical Analysis

Statistical analyses were carried out using SPSS version 28 for Windows (IBM, Armonk, NY, USA). The normality of all variables was tested using the Kolmogorov–Smirnov test. Unless otherwise stated, data are presented as mean and standard deviation (SD). Means and medians are presented for skewed data. Between-group differences at baseline were examined using independent *t*-tests, and 2-way analysis of variance was used to determine the intervention's effect. Paired sample *t*-tests were applied to evaluate within-group pre-to-post performance changes. Effect sizes were calculated by converting partial eta squared values to Cohen's *d* values, classified as small ($0.00 \leq d \leq 0.49$), medium ($0.50 \leq d \leq 0.79$),

and large ($d \geq 0.80$)] [35]. Training-related effects were assessed by 2-way analysis of variance (group × time). If a significant F value was observed, Scheffe's post hoc procedure was applied to locate pairwise differences. The criterion for statistical significance was set at $p < 0.05$ (two-tailed). The reliability of the running, jumping, throwing, and upper and lower limb force tests was assessed using interclass correlation coefficients [36].

**Table 4.** The training program assigned for the contrast strength training group.

|  | Week | % 1RM Heavy Load (Set × Repetition) | % 1RM Light Load (Set × Repetition) | Recovery Time (min) |
|---|---|---|---|---|
| Cycle 1 | 1 | 60 (6 × 6) | 30 (8 × 6) | 1–2 |
|  | 2 | 65 (6 × 6) | 35 (8 × 6) | 1–2 |
|  | 3 | 70 (6 × 5) | 40 (6 × 6) | 1–2 |
|  | 4 | 70 (3 × 5) | 40 (3 × 6) | 1–2 |
| Cycle 2 | 5 | 75 (6 × 4) | 45 (6 × 6) | 1–2 |
|  | 6 | 75 (6 × 4) | 45 (6 × 6) | 1–2 |
|  | 7 | 75 (6 × 4) | 45 (6 × 6) | 1–2 |
|  | 8 | 75 (3 × 4) | 45 (3 × 6) | 1–2 |

## 3. Results

A group × time interaction was observed between the COMB and CONT groups in sprint performance over distances of 20 m ($p = 0.002$; d = 1.93) and 30 m ($p < 0.001$; d = 1.00 (Table 5). A second group × time interaction was found between the COMB and CST groups for the 20 m ($p = 0.042$; d = 0.19) and 30 m ($p = 0.039$; d = 0.13) sprints. However, the CST group showed no significant difference from the COMB and CONT groups for all evaluated sprint distances. The modified T-test displayed only one group × time interaction ($p = 0.021$; d = 1.75; Table 5) between the COMB and CONT groups, suggesting that the COMB group presented the greatest improvement (7.6%). Data from the repeated change of direction test showed that both the COMB and CST groups enhanced their RCOD-Total Time and RCOD-Mean compared to the control group. Indeed, a significant group × time interaction was found in RCOD-Total Time and RCOD-Mean for the COMB and CST groups vs. the CONT group ($p = 0.001$ for both; Table 5). However, no significant group × time interactions were found in RCOD-Best and RCOD-Fatigue Index. For the change of direction deficit (RCOD-Deficit), there were no significant interactions, suggesting that RCOD-Deficit remained unchangeable over time regardless of the moderate changes (−1.46% to 0.64%).

The ball throwing velocities showed a group × time interaction for the COMB vs. CST and CONT groups (3 step running throw: $p = 0.003$, d = 0.08 and $p < 0.001$, d = 1.18; running throw: $p = 0.02$, d = 0.24 and $p = 0.031$, d = 1.22; jumping throw: $p = 0.001$, d = 0.61 and $p < 0.001$, d = 1.11; Table 6). The handgrip strength displayed a group × time interaction between the COMB and CONT groups for right and left hands ($p = 0.004$, d = 0.86 and $p = 0.032$, d = 0.45, respectively). However, the results for the CST group remained statistically unchanged for the ball throwing velocities and handgrip strength.

**Table 5.** Sprinting, agility, and repeated change of direction performance of the three groups after the 8-week intervention.

| Variables | Group | Pre | Post | %Δ | Student's t-Test (p-Value) | Cohen's d | ANOVA Group × Time Interaction |
|---|---|---|---|---|---|---|---|
| | | | | Sprint (s) | | | |
| 5 m | CST | 0.99 ± 0.14 | 0.96 ± 0.10 | 3.16 ± 5.82 | 0.075 | 0.32 | 1.000 ¥ |
| | COMB | 0.96 ± 0.08 | 0.91 ± 0.04 | 5.51 ± 6.01 | 0.014 | 0.95 | 0.150 * |
| | CONT | 0.99 ± 0.06 | 0.99 ± 0.07 | −0.39 ± 2.14 | 0.508 | −0.01 | 0.580 # |
| 20 m | CST | 3.13 ± 0.28 | 3.04 ± 0.22 | 2.51 ± 2.87 | 0.013 | 0.44 | 1.000 ¥ |
| | COMB | 3.00 ± 0.11 | 2.89 ± 0.06 | 3.73 ± 3.52 | 0.005 | 1.93 | 0.002 * |
| | CONT | 3.12 ± 0.18 | 3.15 ± 0.22 | −1.22 ± 5.53 | 0.465 | −0.19 | 0.042 # |

**Table 5.** *Cont.*

| Variables | Group | Pre | Post | %Δ | Student's *t*-Test (*p*-Value) | Cohen's *d* | ANOVA Group × Time Interaction |
|---|---|---|---|---|---|---|---|
| | | | | Sprint (s) | | | |
| 30 m | CST | 4.41 ± 0.30 | 4.34 ± 0.32 | 1.67 ± 2.05 | 0.017 | 0.26 | 0.825 ¥ |
| | COMB | 4.24 ± 0.14 | 4.12 ± 0.09 | 2.57 ± 2.89 | 0.012 | 1.00 | <0.001 * |
| | CONT | 4.50 ± 0.32 | 4.46 ± 0.29 | 0.76 ± 2.36 | 0.263 | 0.13 | 0.039 # |
| | | | | Change of Direction (s) | | | |
| Modified T-Test | CST | 6.06 ±0.38 | 5.77 ± 0.36 | 4.72 ± 1.48 | 0.001 | 0.81 | 0.231 ¥ |
| | COMB | 6.06 ± 0.28 | 5.59 ± 0.27 | 7.62 ± 3.16 | 0.001 | 1.75 | 0.021 * |
| | CONT | 6.15 ± 0.40 | 6.11 ± 0.39 | 0.71 ± 1.49 | 0.121 | 0.12 | 1.000 # |
| RCOD-Best Time | CST | 5.48 ± 0.28 | 5.41 ± 0.27 | 1.28 ± 2.01 | 0.051 | 0.27 | 0.068 ¥ |
| | COMB | 5.54 ± 0.19 | 5.41 ± 0.17 | 2.24 ± 1.26 | 0.001 | 0.73 | 0.202 * |
| | CONT | 5.59 ± 0.24 | 5.66 ± 0.25 | −1.30 ± 1.83 | 0.030 | −0.31 | 1.000 # |
| RCOD-Total Time | CST | 33.8 ± 1.84 | 33.4 ± 1.76 | 1.51 ± 1.82 | 0.016 | 1.30 | 0.001 ¥ |
| | COMB | 33.9 ± 1.24 | 33.2 ± 0.95 | 2.29 ± 1.53 | 0.001 | 0.74 | 0.001 * |
| | CONT | 35.3 ± 1.48 | 35.4 ± 1.33 | −0.30 ± 1.36 | 0.476 | −0.07 | 1.000 # |
| RCOD-Mean Time | CST | 5.64 ± 0.31 | 5.56 ± 0.29 | 1.51 ± 1.82 | 0.016 | 0.30 | 0.001 ¥ |
| | COMB | 5.66 ± 0.21 | 5.52 ± 0.16 | 2.29 ± 1.53 | 0.001 | 0.74 | <0.001 * |
| | CONT | 5.88 ± 0.25 | 5.89 ± 0.22 | −0.30 ± 1.36 | 0.476 | −0.07 | 1.000 # |
| RCOD-Fatigue Index | CST | 3.02 ± 1.41 | 2.79 ± 1.58 | −28.5 ± 70.1 | 0.529 | 0.16 | 0.249 ¥ |
| | COMB | 2.16 ± 1.19 | 2.11 ± 0.85 | −17.9 ± 82.7 | 0.885 | 0.06 | 0.028 * |
| | CONT | 5.34 ± 5.79 | 4.31 ± 5.90 | −57.8 ± 57.6 | 0.004 | 0.18 | 1.000 # |
| COD-Deficit | CST | −2.35 ± 0.31 | −2.36 ± 0.29 | −0.77 ± 6.01 | 0.771 | 0.04 | 0.371 ¥ |
| | COMB | −2.53 ± 0.19 | −2.52 ± 0.15 | 0.19 ± 4.69 | 0.776 | −0.06 | 1.000 * |
| | CONT | −2.47 ± 0.28 | −2.51 ± 0.27 | −1.70 ± 6.89 | 0.478 | 0.14 | 0.090 # |

CST: Contrast strength training group; COMB: Combined isometric and plyometric training group; CONT: Control group; ANOVA: Analysis of Variance; %Δ: percentage difference. ¥: *p*-value between CST and CG groups; *: *p*-value between COMB and CONT groups; #: *p*-value between CST and COMB groups.

Only the CMJ displayed a group × time interaction effect (*p* = 0.014, d = 0.85; Table 7). Our data suggested that the COMB training group improved its CMJ performance more than the CST and CONT groups. However, the SJ and lower limb isometric strength results between the three groups remained similar.

**Table 6.** Throw velocity and handgrip strength for the three groups after the 8-week intervention.

| Variables | Group | Pre | Post | %Δ | Student's *t*-Test (*p*-Value) | Cohen's *d* | ANOVA Interaction (Group × Time) |
|---|---|---|---|---|---|---|---|
| | | | | Ball Throwing Velocity (m/s) | | | |
| 3 Step Running Throw | CST | 20.6 ± 1.49 | 22.1 ± 1.52 | 7.0 ± 2.93 | 0.001 | 1.08 | 1.000 ¥ |
| | COMB | 22.3 ± 1.27 | 23.8 ± 1.23 | 5.9 ± 2.88 | 0.001 | 1.18 | <0.001 * |
| | CONT | 21.4 ± 1.24 | 21.5 ± 1.03 | 0.46 ± 2.21 | 0.517 | 0.08 | 0.003 # |
| Running Throw | CST | 19.4 ± 1.46 | 21.3 ± 1.47 | 8.4 ± 7.65 | 0.003 | 1.32 | 1.000 ¥ |
| | COMB | 21.5 ± 0.98 | 22.9 ± 1.43 | 6.1 ± 4.83 | 0.002 | 1.22 | 0.031 * |
| | CONT | 20.9 ± 1.55 | 21.2 ± 1.25 | 1.6 ± 3.02 | 0.116 | 0.24 | 0.020 # |
| Jumping Throw | CST | 19.1 ± 1.37 | 21.1 ± 1.11 | 9.4 ± 3.01 | 0.001 | 1.65 | 1.000 ¥ |
| | COMB | 21.3 ± 1.24 | 22.6 ± 1.29 | 5.9 ± 2.91 | 0.001 | 1.11 | <0.001 * |
| | CONT | 19.8 ± 1.66 | 20.6 ± 1.08 | 4.0 ± 5.97 | 0.043 | 0.61 | 0.001 # |
| Penalty Throw | CST | 18.4 ±1.33 | 20.3 ± 1.36 | 9.4 ± 1.70 | 0.001 | 1.47 | 1.000 ¥ |
| | COMB | 19.4 ± 1.36 | 21.3 ± 1.34 | 9.2 ± 2.61 | 0.001 | 1.52 | 0.102 * |
| | CONT | 18.8 ± 2.04 | 19.8 ± 1.61 | 4.7 ± 8.59 | 0.095 | 0.54 | 0.160 # |
| | | | | Handgrip Strength (N) | | | |
| Right Hand | CST | 401 ± 62 | 444 ± 65 | 9.6 ± 5.0 | 0.001 | 0.70 | 0.178 ¥ |
| | COMB | 470 ± 66 | 522 ± 59 | 10.2 ± 3.9 | 0.001 | 0.86 | 0.004 * |
| | CONT | 419 ± 89 | 435 ± 78 | 3.9 ± 8.8 | 0.163 | 0.20 | 0.106 # |

**Table 6.** *Cont.*

| Variables | Group | Pre | Post | %Δ | Student's *t*-Test (*p*-Value) | Cohen's *d* | ANOVA Interaction (Group × Time) |
|---|---|---|---|---|---|---|---|
| | | | Handgrip Strength (N) | | | | |
| Left Hand | CST | 388 ± 44 | 432 ± 47 | 10.1 ± 4.11 | 0.001 | 1.00 | 0.079 ¥ |
| | COMB | 428 ± 79 | 462 ± 78 | 7.4 ± 3.94 | 0.001 | 0.45 | 0.032 * |
| | CONT | 378 ± 91 | 397 ± 88 | 4.7 ± 11.4 | 0.186 | 0.23 | 1.000 # |

CST: Contrast strength training group; COMB: Combined isometric and plyometric training group; CONT: Control group; ANOVA: Analysis of Variance; %Δ: percentage difference. Data are presented as mean ± standard deviation. ¥: *p*-value between CST and CG groups; *: *p*-value between COMB and CONT groups; #: *p*-value between CST and COMB groups.

**Table 7.** Countermovement jump, squat jump, and lower limb isometric strength in the three groups after the 8-week intervention.

| Variables | Group | Pre | Post | %Δ | Student's *t*-Test (*p*-Value) | Cohen's *d* | ANOVA Interaction (Group × Time) |
|---|---|---|---|---|---|---|---|
| Countermovement Jump (cm) | CST | 29.5 ± 6.9 | 32.3 ± 7.5 | 8.7 ± 4.1 | 0.001 | 0.41 | 1.000 ¥ |
| | COMB | 33.0 ± 2.9 | 35.5 ± 3.3 | 7.1 ± 2.5 | 0.001 | 0.85 | 0.014 * |
| | CONT | 29.6 ± 4.9 | 30.0 ± 4.5 | 1.1 ± 8.6 | 0.647 | 0.08 | 0.324 # |
| Squat Jump (cm) | CST | 29.2 ± 6.9 | 31.9 ± 7.9 | 8.4 ± 3.7 | 0.001 | 0.39 | 1.000 ¥ |
| | COMB | 31.1 ± 3.3 | 33.6 ± 3.8 | 7.3 ± 4.1 | 0.001 | 0.74 | 0.409 * |
| | CONT | 29.5 ± 4.1 | 29.9 ± 4.5 | 1.17 ± 3.6 | 0.168 | 0.10 | 1.000 # |
| Back Extensor Isometric Strength (N) | CST | 1495 ± 287 | 1603 ± 273 | 6.81 ± 6.36 | 0.003 | 0.40 | 0.117 ¥ |
| | COMB | 1594 ± 217 | 1855 ± 259 | 14.0 ± 3.82 | 0.001 | 1.14 | 0.107 * |
| | CONT | 1494 ± 332 | 1536 ± 304 | 3.08 ± 5.58 | 0.076 | 0.14 | 1.000 # |

CST: Contrast strength training group; COMB: Combined isometric and plyometric training group; CONT: Control group; ANOVA: Analysis of Variance; %Δ: percentage difference. Data are presented as mean ± standard deviation. ¥: *p*-value between CST and CONT groups; *: *p*-value between COMB and CONT groups; #: *p*-value between CST and COMB groups.

## 4. Discussion

This study compared the effects of 8 weeks of contrast strength training and combined isometric and plyometric training on physical performance measures in elite junior handball players. The COMB approach yielded greater enhancements in sprinting, 3 step running throw, running throw, and jumping performance compared to the CST group. Both training strategies significantly improved most neuromuscular performance measures compared to the CONT group. These results indicate that alternating between isometric and plyometric exercises in the same session holds significant promise as a potential strategy to enhance explosive action in male handball players.

We expected that training combining both moderate-intensity isometric and light-load, high-velocity plyometric drills would result in performance increments in handball-related tasks such as jumping, sprinting, and throwing. The static nature of muscular contraction in isometric mode is often perceived as less beneficial for dynamic sports performance [22]. Indeed, dynamic strength training is generally considered the preferred mode of strength training due to its ability to improve sports-related dynamic performance [37]. Therefore, it was not surprising that both the COMB and CST groups substantially improved their single effort sprint, modified T-test, and jumping test performance as both training programs included static or 'heavy' loading and higher velocity dynamic movements. However, the improvements in these tasks were generally greater following COMB than CST. This finding is easily explained by training specificity [38], as only the COMB training program included sprinting, jumping, and change of direction exercises. Interestingly, the between-group differences in the repeated change of direction test results were unclear, with the COMB group coming out on top for the best single trial (d = 0.73 vs. d = 0.27) and the CST group showing greater improvement for total time (d = 0.74 vs. d = 1.30). The reasons for these findings are hard to determine, although we can speculate that the COMB group's greater

improvement in best time could be due to the inclusion of specific change of direction drills, while the CST group doubling up on resistance training movements may have had a beneficial effect on muscular endurance, leading to more consistent times over the six COD repetitions.

Based on the training specificity concept, one might have expected the COMB group to outperform the CST group when examining the results of the throwing tests in the present study. However, this was not the case, as the between-group differences were unclear or minimal. While the COMB training program did include explosive movements, it did not have actual throwing or shooting exercises or drills. Therefore, authentic 'specific' throw training was not included for either group, suggesting that the large improvements in the COMB (d = 1.11–1.52) and CST (d = 1.08–1.64) groups were predominantly due to improving full-body strength and power. In line with our findings, a 12-week contrast training program effectively improved soccer-specific skills such as vertical jumping, sprinting, and kicking speed in 30 U16 soccer players [21]. Furthermore, tasks involving the stretch-shortening cycle (e.g., plyometrics) use the elastic properties of connective tissue and muscle fibers. During the deceleration or negative phase of the movement, the muscle accumulates elastic energy, which is then released during the acceleration or positive phase to enhance the force and power output of the muscle [39]. This pattern of muscle contractions during the stretch-shortening cycle almost certainly benefits sports performance, such as acceleration, change of direction, throwing, and vertical and horizontal jumping [8,12]. Finally, throwing was the only category where the CONT group experienced notable improvements (d = 0.08–0.61), suggesting that a learning effect may have contributed to the lack of differences between the COMB and CST groups.

Strength is often an important underpinning of improved dynamic and sports performance [37]. While both experimental groups showed improved isometric strength, the COMB group (d = 1.14) improved by a greater magnitude than the CST group (d = 0.40). Since shifting the force portion of the power equation (mechanical power = force × velocity) is known to improve dynamic performance, the greater improvement in strength is a plausible explanation for the COMB group experiencing greater magnitudes of improvement in the single effort sprint (d = 0.95–1.93 vs. d = 0.26–0.44), jump (d = 0.74–0.85 vs. d = 0.39–0.41), and change of direction T-test (d = 1.75 vs. d = 0.81) tasks compared to the CST group. However, the reason behind the greater strength increase in the COMB group is hard to determine as the isometric and 'heavy load' exercises were prescribed at the same relative loads for the COMB and CST groups, respectively. Nevertheless, it is possible that the total time was greater for the COMB group. Another potential explanation is that plyometrics in isolation can effectively improve maximal strength due to high mechanical and neural stress levels while unloaded. Indeed, a meta-analysis determined that plyometrics can substantially improve dynamic and static strength assessments (Hedges' g = 0.57–1.11) [40]. While the CST training program also incorporated higher velocity movements, the velocity achieved during low-load resistance training was still far slower than during the sprinting and plyometric exercises [41]. Therefore, the general superiority of COMB over CST could be attributed to the aforementioned factors.

This work had limitations that should be considered. Indeed, the study did not incorporate electromyography measurements, which could have provided further insight into muscle activation patterns and the neuromuscular response during the training interventions. Additionally, we compared the combined isometric and plyometric training program with a contrast training program, not with more traditional training methods. Therefore, we cannot say the COMB program is superior to other training approaches. Likewise, the isometric contractions in the present study were performed at submaximal intensities, begging the question of whether similar results would have occurred with intensities over the proxy movements' 1RM. In addition, to fortify the study's impact, future research should consider a longer follow-up period and expand the sample size to enhance the study's generalizability. Furthermore, incorporating female participants in subsequent studies could provide valuable insight into gender-specific responses to

training interventions. Finally, our findings may not have been similar in other age groups, sporting backgrounds, or resistance training-naïve athletes.

## 5. Conclusions

Incorporating an 8-week, biweekly, upper and lower body contrast training or combined isometric and plyometric training regimen into elite junior handball players' regular handball training schedule enhanced markers of handball-related performance. The isometric and plyometric training regimen was more effective in enhancing 20 m and 30 m sprint, running throw, 3 step running throw, and jumping throw performance. These findings suggest that coaches should consider integrating isometric and plyometric elements into in-season resistance training sessions for elite junior male handball players.

**Author Contributions:** Conceptualization, M.S.C. and H.A.; methodology, M.S.C. and H.A.; software, M.S.C., H.A., A.K. and O.T.; validation, M.S.C., A.K. and H.A.; formal analysis, H.A. and O.T.; investigation, H.A. and A.K.; resources, M.S.C.; data curation, H.A. and A.K.; writing—original draft preparation, H.A., O.T. and A.K.; writing—review and editing, M.S.C., D.J.O. and R.S.; visualization, R.S.; supervision, M.S.C.; project administration, H.A. and M.S.C.; funding acquisition, R.S. All authors have read and agreed to the published version of the manuscript.

**Funding:** This research was funded by Ministry of Higher Education and Scientific Research, Tunis, Tunisia. We acknowledge the financial support of the Open Access Publication Fund of the Martin-Luther-University Halle-Wittenberg.

**Institutional Review Board Statement:** The study was conducted according to the guidelines of the Declaration of Helsinki, according to current national laws and regulations, and approved by the Institutional Review Board of Research Laboratory (LR23JS01) Sport Performance, Health & Society, Higher Institute of Sport and Physical Education of Ksar-Saîd, University of Manouba.

**Informed Consent Statement:** Informed consent was obtained from all subjects involved in the study.

**Data Availability Statement:** The raw data supporting the conclusions of this article will be made available by the authors without undue reservation.

**Acknowledgments:** The authors thank the Ministry of Higher Education and Scientific Research, Tunis, Tunisia, for financial support.

**Conflicts of Interest:** The authors declare no conflict of interest.

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
