# Peer review of "The Effect of Combined Isometric and Plyometric Training versus Contrast Strength Training on Physical Performance in Male Junior Handball Players"

_applsci, doi:10.3390/app13169069_

Round 1

Reviewer 1 Report

In the introduction, there are two statements made in a way as to sound like absolute fact, while are still hypotheses, and as such should be lightened or at least made non-absolute [48-49] [59-60]. In the conclusions it is made clear that it is just a paraphrasing issue and not methodological.

Quotes of a direct word-per-word citation are mandatory, citation is not sufficient, falls within the risk of plagiarism [68-70].

[232] "ball" word is missing in exercise description.

Overall commendable work.

[ ] - refers to the lines in the draft

Reviewer 2 Report

The study compared the effects of combine isometric and plyometric training with contrast training. Both training concept have garnered much interest from both academics and practitioners in recent years. The study was generally well designed and manuscript well writtened. 

I only have a few feedbacks that i hope the authors can address.

Line 82-85: “However, recent …………stiffness, and angle-specific strength”. Consider including a couple more studies showing the benefits of including isometric training. There are a number of studies published in the past few years showing the benefits of isometric training on sports performance. This will provide more support for your statement.

Line 146-153: How was timing measured for the modified T-test?

Line 213-232: What was the rest time between isometric and plyometric exercises? Were the isometric and plyometric  performed as contrast pairs or did participants completed all isometric sets before performing plyometric sets?

Line 238: “All sets and all repetitions were separated by 240 s of passive recovery” you may want to remove “repetitions”. It sounds like participants had to rest for 240s after each repetition as well.

What was the rest period between heavy and light sets? Or was it immediate with no rest?

Also, how was the concentric and eccentric phase of CST performed? i.e. as fast as possible for both phases; fast conc but slow ecc, etc.

Line 136-208: Please provide ICC and %CV data for each tested variable. Either insert in text or display in table.

Reviewer 3 Report

I would like to read a paragraph about the training program that the Control group followed during the test period (lines 119-122).

During the test period did the authors had any drop-outs from the participants? If yes, how they treat it, and what was the effect on the results of the study? 

Reviewer 4 Report

The authors effectively addressed the research question by conducting a well-designed 8-week intervention study with a randomized controlled trial approach. The inclusion of a control group adds to the rigor of the study, allowing for clearer comparisons between the intervention groups and the control group.

The results of this study demonstrated significant improvements in various performance parameters among participants in the combined isometric and plyometric training group (COMB). Notably, the enhancements in change-of-direction abilities, sprinting performance, and jumping skills stand out, indicating the potential benefits of this training approach for handball players.

However, to strengthen the study's impact, I recommend considering a longer follow-up period and expanding the sample size to enhance the study's generalizability. Additionally, including female participants in future studies could provide valuable insights into gender-specific responses to the training interventions.
